# Gene expression and chromatin conformation of microglia in virally suppressed people with HIV

Johannes CM Schlachetzki[1,2,*], Sara Gianella[3,*], Zhengyu Ouyang[1,*], Addison J Lana[1], Xiaoxu Yang[4], Sydney O'Brien[1], Jean F Challacombe[1], Peter J Gaskill[5], Kelly L Jordan-Sciutto[6], Antoine Chaillon[3], David Moore[7], Cristian L Achim[8], Ronald J Ellis[2], Davey M Smith[3], Christopher K Glass[1]

The presence of HIV in sequestered reservoirs is a central impediment to a functional cure, allowing HIV to persist despite life-long antiretroviral therapy (ART), and driving a variety of comorbid conditions. Our understanding of the latent HIV reservoir in the central nervous system is incomplete, because of difficulties in accessing human central nervous system tissues. Microglia contribute to HIV reservoirs, but the molecular phenotype of HIV-infected microglia is poorly understood. We leveraged the unique "Last Gift" rapid autopsy program, in which people with HIV are closely followed until days or even hours before death. Microglial populations were heterogeneous regarding their gene expression profiles but showed similar chromatin accessibility landscapes. Despite ART, we detected occasional microglia containing cell-associated HIV RNA and HIV DNA integrated into open regions of the host's genome (~0.5%). Microglia with detectable HIV RNA showed an inflammatory phenotype. These results demonstrate a distinct myeloid cell reservoir in the brains of people with HIV despite suppressive ART. Strategies for curing HIV and neurocognitive impairment will need to consider the myeloid compartment to be successful.

## Introduction

Triggered by the tremendous advances in antiretroviral therapy (ART), HIV has become a chronic disease with life expectancy nearly comparable to the general population (Deeks et al, 2013). But even with ART, ~40–60% of people with HIV develop neurocognitive impairment during their lifetime (Heaton et al, 2010). Despite ART, the brain acts as a sanctuary for HIV (Price et al, 1988), as HIV RNA and HIV DNA are often detected in the brain

tissue and cerebrospinal fluid (CSF) of people with HIV with viral suppression in plasma (Gianella et al, 2016; Mohammadzadeh et al, 2021). The virus's capacity to persist indefinitely in specific long-lived cell populations in anatomical sanctuaries like the brain despite long-term ART is the main barrier to an HIV cure. Within the central nervous system (CNS), microglia are thought to be the main target of HIV infection and might contribute to viral rebound once ART is interrupted (Wallet et al, 2019). Microglia infected with HIV may drive chronic low-grade inflammation and persistent low-level virus replication, contributing to synaptodendritic injury (Kaul et al, 2001; Ellis et al, 2007, 2023; Saylor et al, 2016).

Recent advances in single-cell technologies demonstrate that microglia are not a homogeneous cell population but highly diverse and plastic cells that are highly responsive to their microenvironment (Matsudaira & Prinz, 2022). The generation of single-cell and single-nucleus transcriptomic data from isolated human microglia from neurotypical and diseased brains has revealed multiple, small subclusters—microglial states—as characterized by the up-regulation of distinct marker genes compared with homeostatic microglia (Sankowski et al, 2019; Olah et al, 2020). Homeostatic microglial marker genes include surface receptors such as *CX3CR1*, *P2RY12*, *TMEM119*, and *SALL1*, central regulators of microglial identity (Gosselin et al, 2017; Fixsen et al, 2023). The up-regulation of MHCII genes such as *CD74* and *HLA-DRA* indicates that microglia participate in antigen presentation. Other microglial states include interferon-responsive microglia (e.g., *IFITM3*, *IFIT1*, *IFIT3*, *ISG15*), inflammatory microglia (e.g., *CCL2*, *CCL3*, *CCL4*), proliferative microglia (e.g., *MKI67*, *PCNA*), and a small subset reminiscent of mouse disease–associated microglia (e.g., *APOE*, *LPL*) (Sankowski et al, 2019). Although HIV DNA has been detected in microglia isolated by laser microdissection and by in situ hybridization in human postmortem studies (Trillo-Pazos et al, 2003; Thompson et al, 2011; Ko et al, 2019), the molecular profile of HIV-

[1]Department of Cellular and Molecular Medicine, University of California San Diego, San Diego, CA, USA  [2]Department of Neurosciences, University of California San Diego, San Diego, CA, USA  [3]Department of Medicine, Division of Infectious Diseases and Global Public Health, University of California San Diego, San Diego, CA, USA  [4]Department of Human Genetics, University of Utah, Salt Lake City, UT, USA  [5]Department of Pharmacology and Physiology, Drexel University College of Medicine, Philadelphia, PA, USA  [6]Department of Oral Medicine, School of Dental Medicine, University of Pennsylvania, Philadelphia, PA, USA  [7]Department of Psychiatry, University of California San Diego, San Diego, CA, USA  [8]Department of Pathology, University of California San Diego, San Diego, CA, USA

Correspondence: jschlachetzki@ucsd.edu; ckg@ucsd.edu
*Johannes CM Schlachetzki, Sara Gianella, and Zhengyu Ouyang contributed equally to this work

infected human microglia in the brain of virally suppressed people with HIV is poorly understood (Schlachetzki et al, 2022).

Some of the major open questions about the latent HIV reservoir in the CNS concern the extent of the reservoir in this compartment and the gene expression profile of HIV-infected microglia in people with HIV on suppressive ART. Several technical roadblocks needed to be overcome to answer these questions, including (i) difficulty in accessing the brain in living people with HIV; (ii) that microglia represent only ~5% of all brain cell types; (iii) that myeloid cell isolation requires fresh, non-archived viable tissue, requiring rapid autopsy procedures to minimize the postmortem interval; (iv) that participants are often not well characterized close to the time of death in terms of monitoring viral load and ART intake; and (v) that no known viral protein or unambiguous cell surface marker currently enables the detection of HIV in quiescent cells, and long-lived latently infected cells are largely transcriptionally silent.

To overcome these roadblocks, we took advantage of the "Last Gift" program at the University of California San Diego (Perry et al, 2020; Rawlings et al, 2021). The Last Gift program provides a unique opportunity to advance HIV cure research by enrolling altruistic people with HIV who are diagnosed with a terminal illness from a non–HIV-related condition and who donate their tissues and cells for a rapid research autopsy to advance HIV cure research (Chaillon et al, 2020; Rawlings et al, 2021). Thus, we were able to isolate high-quality microglia from the dorsolateral prefrontal cortex (DLPFC) of these people with HIV within a postmortem interval of less than 4 h. Using this tissue, we performed single-cell RNA sequencing (scRNA-seq) and single-cell assay for transposase-accessible chromatin with sequencing (scATAC-seq) from three individuals on sustained ART to (1) establish an atlas of the gene expression profile and active gene regulatory element; (2) detect HIV RNA and HIV DNA in the myeloid compartment; and (3) uncover the gene expression signature and chromatin conformation of HIV-infected microglia.

# Results

## Study participants and HIV reservoir quantification

We enrolled three male participants with chronic HIV infection on long-standing ART who participated in the Last Gift program at the University of California San Diego (Fig 1A and Table 1). Specifically, LG05 was a 57-yr-old male with a 23-yr history of HIV and was diagnosed with amyotrophic lateral sclerosis. LG14 died of a squamous cell cancer of the tongue at age 59, 14 yr after acquiring HIV. LG15 was the oldest individual at 89 yr old with a 20-yr history of HIV, who died of heart failure. All three people with HIV had been treated with an antiretroviral drug regimen consisting of nucleoside/nucleotide reverse transcriptase inhibitors, a non-nucleoside reverse transcriptase inhibitor, and an integrase inhibitor until at least 2 wk before death. Levels of total HIV DNA from the bulk DLPFC were measured by droplet digital (dd)PCR. For LG05 and LG15, detectable HIV DNA levels were low (11 and 2 HIV DNA copies per $10^6$ cells, respectively). LG14 showed the highest levels of HIV DNA with 258 copies per $10^6$ cells (Fig 1B).

## Single-cell transcriptome of CD45$^+$ cells in the prefrontal cortex of people with HIV

We focused on the myeloid cell compartment of the DLPFC from each individual. Specimens from the DLPFC were received within a postmortem interval of 4 h. Approximately 500 mg of the DLPFC tissue was dissociated at 4°C without using enzymes to avoid artificial activation of microglia and other immune cells (Gosselin et al, 2017; Marsh et al, 2022). We isolated CD45$^+$ cells via FACS (Figs 1C and S1). To assess the transcriptomic profiles of CD45$^+$ cells, we performed single-cell RNA-seq using the microfluidic-based 10X Chromium platform, which allows profiling of 3'-polyadenylated RNAs (Zheng et al, 2017). In total, we sequenced 30,380 CD45$^+$ cells, and after quality control and removal of cells with >20% mitochondrial content, we profiled 25,091 cells obtained from all three individuals (Fig S2A and B). After embedding the datasets into the nearest neighbor space using Uniform Manifold Approximation and Projection, we detected 15 distinct clusters of CD45$^+$ cells (Fig 1D). Our sorting strategy included all CD45$^+$ cells from the three DLPFCs and generated predominantly cells expressing *PTPRC* (encoding CD45) and *AIF1* (encoding IBA1) (Fig S2C). Marker gene analysis showed that Cluster 14 (~1.3%) mainly contained lymphoid cells based on marker genes *GZMA*, *GZMB*, *CD3E*, and *CD3G* (Fig S2D and E). Cluster 11 contained a mixture of monocytes and neutrophils (~2.3%) as shown by the expression of marker genes *FCN1*, *ANXA2*, *IL1R2*, *LGALS1*, and *LYZ* (Fig S2D and F). We then focused our analyses on the brain myeloid cell populations.

All brain myeloid cells expressed to varying degrees *CX3CR1*, *P2RY12*, *TREM2*, *GPR34*, and *CSF1R* (Fig 1E and F). Based on marker genes, we annotated 12 microglial cell states and a cluster indicative of perivascular macrophages. The high expression of the *CD163*, *F13A1*, *LYVE1*, and *MRC1* genes in Cluster 13 suggests that these cells were most likely perivascular macrophages (Fig S2G). *SALL1*, a marker distinctly expressed by microglia, was not detected in Cluster 14, suggesting that these cells were perivascular macrophages (Fig S2H). We defined clusters 00, 01, 05, 10, and 12 as homeostatic microglia because they showed the highest expression levels of homeostatic microglial genes such as *CX3CR1*, *P2RY12*, *CSF1*, and *GPR34* (Fig 1E and F). Further analysis showed that clusters 02 and 08 expressed marker genes enriched in mitochondria (e.g., *MT-ATP6*, *MT-COS2*, *MT-CYB*, *MT-ND3*; Supplemental Data 1) and ribosomes (e.g., *RPLP1*, *RPLP13*; Fig S2I). Cells in Cluster 09 were driven by genes driving cell-cycle progression (e.g., *MKI67*, *TOP2A*, *STMNI1*) suggesting microglia undergoing proliferation (Fig S2J and K).

Cluster 03 showed marker genes for *IFI27*, *ISG15*, *APOE*, and *IFITM3* indicating a microglial phenotype involved in antigen presentation and interferon signaling (Fig 2A and B) as indicated by gene enrichment analysis (Fig 2C and Supplemental Data 1). Cluster 04 was defined by genes related to a stress response, including the AP-1 family members *JUN*, *FOS*, and *FOSB* but also *KLF6*, *EGR1*, and heat-shock proteins such as *HSPA1B*. Pro-inflammatory genes such as IL1B and markers associated with senescence were present in Cluster 06. Marker genes for Cluster 07 included genes related to chemokines and cytokine signaling (*CCL3/4*, *CCL4L2*). Cluster 10 was defined by genes involved in the *HIF1* transcription factor pathway, which has been implicated in the altered immunometabolic state of macrophages.

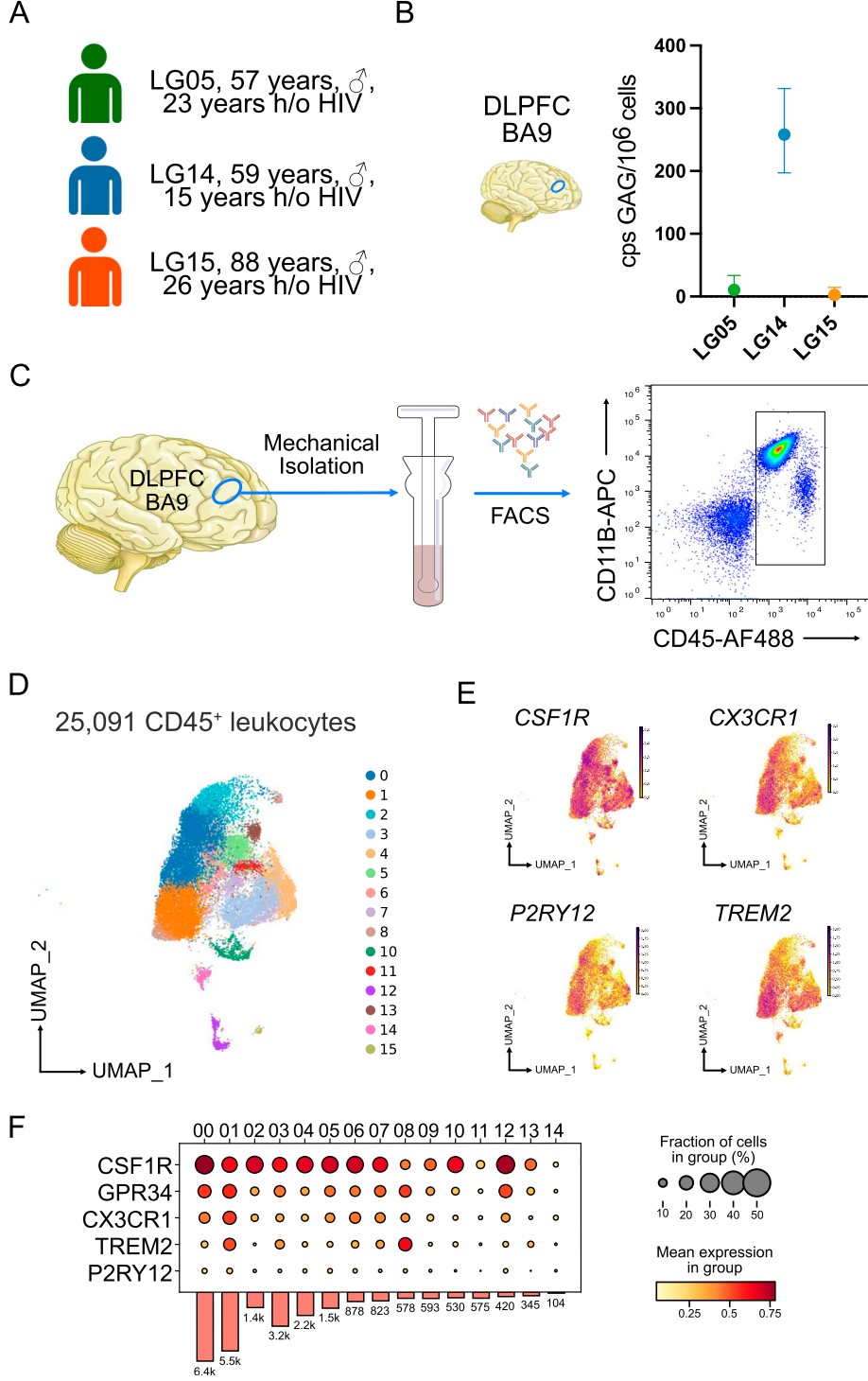

**Figure 1. Experimental design and single-cell transcriptomics of CD45⁺ cells isolated from the cortex of individuals with HIV under antiretroviral therapy.**
**(A)** Clinical cohort. **(B)** Measured GAG copies per million cells from three replicates with the dorsolateral prefrontal cortex using droplet digital PCR. **(C)** Diagrammatic representation of the experimental design. The dorsolateral prefrontal cortex tissue was homogenized followed by gentle mechanical dissociation. After antibody staining, CD45⁺ cells were sorted and processed for downstream applications. **(D)** Uniform Manifold Approximation and Projection for Dimension Reduction of 25,091 profiled CD45⁺ cells isolated from three male Last Gift study participants that passed quality selection criteria. **(E)** Feature plots showing the expression of microglial genes *CX3CR1*, *P2RY12*, *CSF1R*, and *TREM2*. **(F)** Stacked violin plot of homeostatic microglial genes. The color scale depicts the mean expression level. Source data are available for this figure.

## Chromatin accessibility in brain myeloid cells of individuals with HIV

To expand our understanding of the epigenetic regulation of microglia in the context of HIV, we assessed the chromatin accessibility landscape and active gene regulatory networks of human CD45⁺ cells isolated from the DLPFC of three Last Gift participants on ART using the 10X Genomics single-cell ATAC-seq platform (Cusanovich et al, 2015). We sequenced a combined 27,229 CD45⁺ cells isolated from the DLPFC of these three Last Gift participants with a sequencing depth ranging between 120,977,750 and 127,143,163 reads and aligned them to the human reference hg38

**Table 1. Clinical demographics.**

| ID | Sex | Age of death | Duration from HIV diagnosis | ART | Clinical diagnoses |
|---|---|---|---|---|---|
| LG05 | M | 57 | 23 | TAF/FTC/RPV+DTG | ALS |
| LG14 | M | 59 | 15 | FTC/TAF+DTG | SCC of the tongue |
| LG15 | M | 88 | 26 | 3TC+DTG | Multiple medical comorbidities |

Abbreviations: TAF, tenofovir alafenamide (NRTI); FTC, emtricitabine (NRTI), RPV, rilpivirine (NNRTI); DTG, dolutegravir (INSTI); 3TC, lamivudine (NRTI); NRTI, nucleoside/nucleotide reverse transcriptase inhibitors; NNRTI, non-nucleoside reverse transcriptase inhibitors; INSTI, integrase inhibitors; ALS, amyotrophic lateral sclerosis; SCC, squamous cell cancer; ART, antiretroviral therapy.

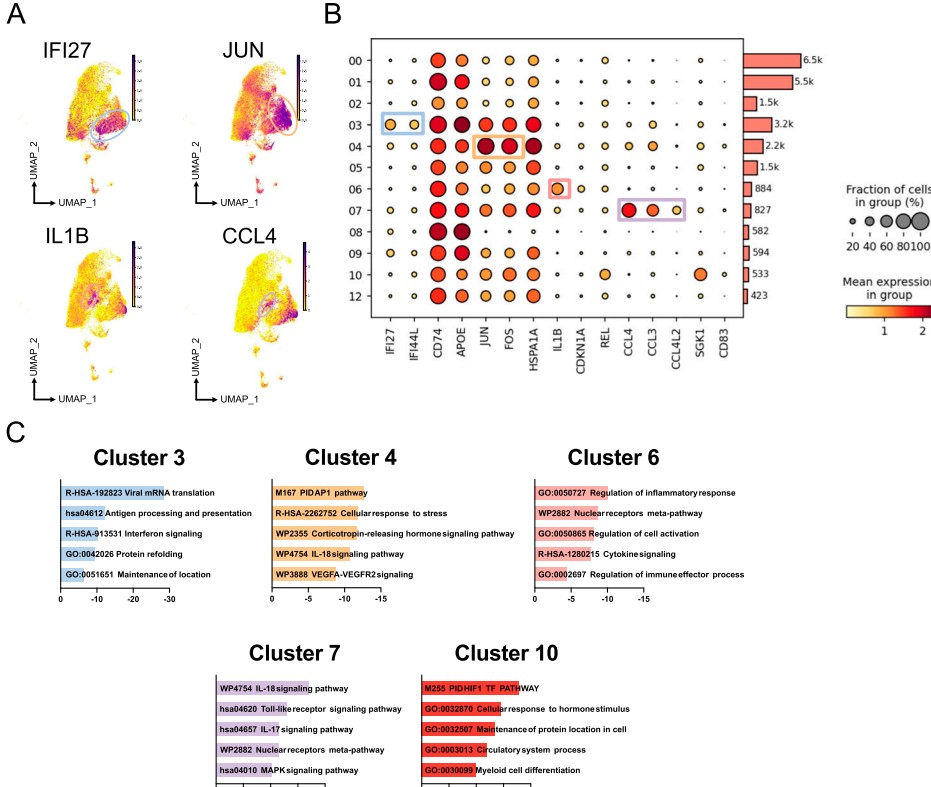

**Figure 2. Microglial phenotypes based on gene expression in the cortex of individuals with HIV under antiretroviral therapy.**
**(A)** Feature plots highlighting the expression of *IFI27* (Cluster 3), *JUN* (Cluster 4), *IL1B* (Cluster 6), and *CCL4* (Cluster 7). **(B)** Dot plot depicting marker genes for each cluster. The size of the circles shows the fraction of cells expressing the genes, and the color scale provides the mean expression level per group. **(C)** Gene ontology enrichment analysis of marker genes for clusters 3, 4, 6, 7, and 11 using Metascape.
Source data are available for this figure.

genome using CellRanger. For data analysis, we used the ArchR package and performed Harmony batch corrections (Korsunsky et al, 2019; Granja et al, 2021). Analysis using Seurat generated 12 clusters, which we assigned to cell types using chromatin accessibility at the promoter as a proxy for gene activity (Fig 3A). We found that 12 clusters showed increased chromatin accessibility at promoters of microglial signature genes such as *CSF1R* and *SALL1* (Figs 3B and S3A) overlapping with ATAC-seq peaks. De novo DNA binding motif enrichment analysis of ATAC-peak in total microglia and subclusters using HOMER (Heinz et al, 2010) showed motifs recognized by microglial lineage–determining factors. These included PU.1, PU.1/IRF ternary complexes, and members of the CTCF, RUNX, C/EBP, MAF, and USF family in line with our previously published epigenetic data on human microglia (Fig S3B) (Gosselin et al, 2017; Nott et al, 2019). To determine whether the ATAC-seq peaks overlap with active gene regulatory elements as marked by acetylation of

lysine residue 27 on histone 3 (H3K27ac), we isolated PU.1+ nuclei from the frozen tissue of the same Last Gift participants and performed H3K27ac ChIP-seq (Fig 3C). Examples for the chromatin accessibility and active gene regulatory elements for microglial signature genes including SALL1, CD4, and TREM2 are given in Fig 3D. We found highly significant enrichment for motifs assigned to lineage-determining transcription factors Sfp1, PU.1:IRF:8, and RUNX1 (Fig 3E). However, we also observed signal-dependent transcription factor families such as AP-1, indicative of reactive microglia.

In contrast to the transcriptome, we were not able to identify distinct cellular states based on chromatin accessibility because of a high degree of similarity between microglia in different clusters. Variation in chromatin accessibility was greater between individuals than in each individual (Fig S3C). Although clusters 12, 13, and 5 mostly came from LG14 and LG15, clusters 06 and 11 contained predominantly cells from donor LG14 (Fig S3D). Most cells in clusters

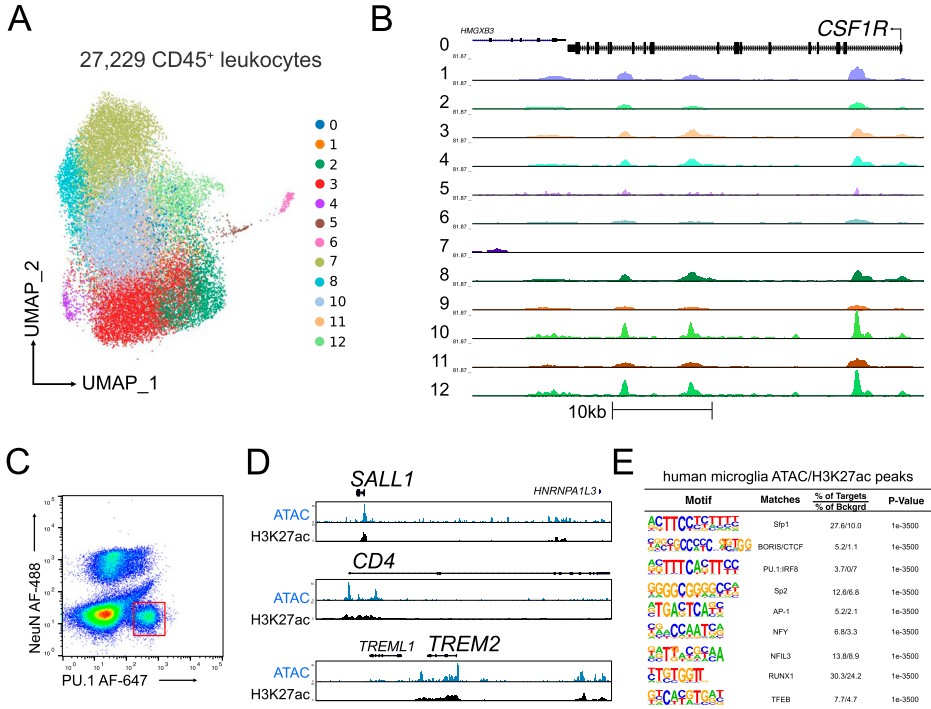

**Figure 3. Chromatin accessibility landscape of microglia isolated from the cortex of people with HIV under antiretroviral therapy identified by single-cell ATAC-seq.**
**(A)** Uniform Manifold Approximation and Projection for Dimension Reduction of 27,229 CD45⁺ cells isolated from three male Last Gift study participants that passed quality selection criteria. **(B)** Genome browser track of all clusters at the *CSF1R* locus. **(C)** Sorting of PU.1+ nuclei from the frozen brain tissue using fluorescence-activated nuclear sorting. **(D)** Genome browser track of ATAC-seq and H3K27ac at the *SALL1*, *CD4*, and *TREM2* loci. **(E)** De novo motif analysis of H3K27ac ChIP-seq peaks anchored in ATAC-seq peaks.

07, 08, 09, and 10 originated from LG5. The high interindividual variability in the chromatin accessibility landscape identified by the single-cell ATAC-seq prevented us from identifying a microglial state–specific chromatin accessibility landscape.

## HIV RNA and HIV DNA detection in brain myeloid cells

After establishing the transcriptomic and chromatin accessibility landscapes of microglia and macrophages from the DLPFC, we evaluated rates of detection of HIV RNA and HIV DNA based on our single-cell RNA-seq and single-cell ATAC-seq datasets. To do this, we aligned the reads obtained from single-cell RNA-seq to the HIV reference genome (HXB2, GenBank K03455.1). HIV RNA was detected in 107 out of 25,091 CD45⁺ cells (0.43%). Of those 107 cells, 99 were in defined microglial populations (Fig 4A). HIV RNA⁺ cells were not evenly distributed across the microglial clusters (Fig 4B) with clusters 3, 4, and 7 showing the highest proportion of HIV RNA⁺ cells (54% of all HIV RNA⁺ cells). Increased interferon signaling and cytokine production and up-regulation of proteins such as the AP-1 associated with stress response are associated with these three clusters.

The number of reads mapping to the HIV reference genome ranged between 1 and 270 per cell. LG14, the individual with the highest levels of HIV DNA from the bulk DLPFC based on ddPCR, also showed the most HIV RNA⁺ cells with 82 cells out of 6,014 sequenced cells or 1.36% (Fig 4C). LG15 had 25 cells with detectable HIV RNA out of 11,978 sequenced cells or 0.2%. No reads mapped to the HIV genome for LG5. We then mapped the reads obtained from single-cell ATAC-seq onto the HIV reference genome. Of 27,299 CD45⁺ cells, we detected HIV DNA integrated into the open regions of the host genome in 80 microglia, indicating integration of the HIV genome into accessible regions of the host genome (Fig 4D). We did not observe an enrichment of microglia with detectable HIV DNA in distinct microglial clusters. In line with the results from scRNA-seq, LG14 had with 64 the greatest number of cells with detectable HIV DNA in open chromatin states, whereas 16 microglia isolated from LG15 had detectable HIV DNA by ATAC-seq (0.54% and 0.18%, respectively). No viral DNA was detected in open chromatin regions in LG05.

To determine whether cells with detectable HIV RNA showed a distinct gene expression pattern relative to cells negative for HIV transcripts, we compared the 107 HIV RNA⁺ cells with 110 randomly selected cells with no detectable HIV RNA and performed permutation analysis (Fig 4E and Supplemental Data 2). Genes that were significantly up-regulated in cells positive for HIV RNA included chemokines *CCL8*, *CXCL8*, *CCL4*, and *CCL3*. Interestingly, CCL8, CCL4, and CCL3 are all ligands for CCR5 and have been shown to suppress HIV infection. HIV RNA⁺ cells also exhibited enrichment for interferon-responsive genes. Specifically, *IFI27*, *IFI44L*, and *IFI6* mRNAs were increased, which are up-regulated during HIV infection in the blood (Mackelprang et al, 2023). The expression of surface markers such as *CD69* and *CD83* was also increased. CD83 plays a critical role in controlling immune responses (Sinner et al, 2023). Interestingly, genes belonging to the AP-1 and KLF transcription factor families also showed increased gene expression. KLF2 is known to repress HIV transcription in T cells (Pedro et al, 2021). HIV has been shown to stimulate AP-1 signaling in macrophages and T cells (Varin et al, 2005; Hokello et al, 2021). Differentially expressed genes, as determined by an adjusted *P*-value of less than 0.05, showed enrichment for GO terms "response to stimulus," "cellular process," "interspecies interaction," and "viral process" (Fig 4F).

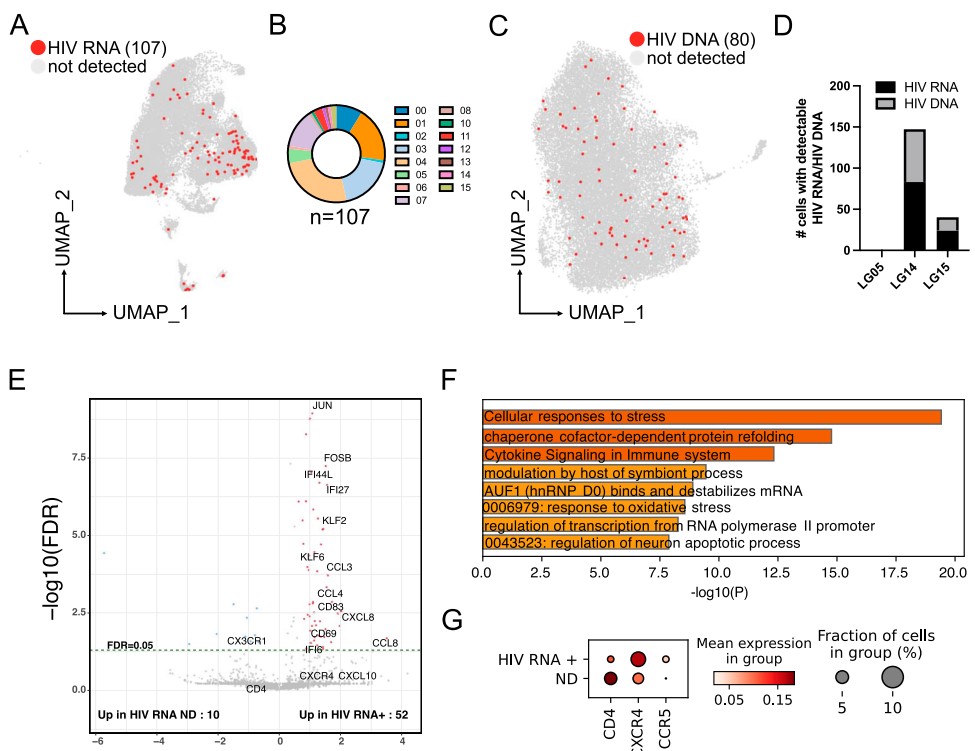

**Figure 4. Differential gene expression of HIV-infected microglial cells.**
**(A)** Uniform Manifold Approximation and Projection for Dimension Reduction presentation of the single-cell RNA-seq dataset highlights cells with detected HIV RNA. **(B)** Pie chart showing the number of cells with detectable HIV RNA per cluster. **(C)** Uniform Manifold Approximation and Projection of single-cell ATAC-seq depicting cells with detectable integrated HIV DNA into the host's genome in regions of open chromatin. **(D)** Number of cells with detectable HIV RNA and HIV DNA per Last Gift participant. **(E)** Volcano plot showing differential gene expression between HIV RNA⁺ cells and random cells with no detectable HIV transcripts. **(F)** Gene ontology analysis of expression of genes up-regulated in microglia with detectable HIV RNA. **(G)** Dot plot depicting the fraction of cells expressing the genes and the mean expression level in cells with detectable (HIV RNA⁺) and non-detectable (ND) HIV RNA.
Source data are available for this figure.

Lastly, *CD4* mRNA was decreased in microglia with detectable HIV RNA, whereas *CCR5* and *CXCR4* mRNAs were up-regulated in HIV RNA–positive microglia, indicating a differential activation state of HIV-infected microglia (Fig 4G).

### Mapping of HIV RNA and HIV DNA

Next, we aligned reads from single-cell RNA-seq and single-cell ATAC-seq of CD45⁺ cells to the HIV reference genome (HXB2, GenBank K03455.1). All reads mapped to the HIV genome were labeled based on the mapping score (MAPQ value) or Phred quality. MAPQ = 3 (=Q3) means the probability of the mapped genome location is 50%, whereas MAPQ > 30 (>Q30) means the probability of the mapped location is larger than 99.9% or incorrectly called in 1 out of 1,000 times. Because part of the reference sequence of the 5′UTR and 3′LTR is identical, we detected reads that mapped ambiguously to these loci. We also observed that most of the reads (>99%) belonged to those two categories, meaning the overall mapping quality was high. In addition, we cross-checked all identified reads that aligned to the HIV reference genome with BLAST and viruSITE to limit false-positive detection of HIV-1 (Stano et al, 2016).

Analysis of the single-cell RNA-seq data revealed a total of 6,504 reads aligned to the HIV reference genome (Fig 5A). We identified two major clusters of sequences: the 3,407 (loci < 1 kb) reads mapped onto the 5′UTR and gag region of the HIV-1 genome, and in contrast, 721 (2 kb < loci < 3 kb) reads between gag and pol. A similar distribution of mapped reads to the HIV genome was previously

shown in T cells isolated from people with HIV (Wei et al, 2023) and in exosomes isolated from HIV-infected myeloid and lymphoid cell lines (Barclay et al, 2017).

Next, we quantified the reads derived from single-cell ATAC-seq and detected 1,123 reads that mapped onto the HIV reference genome (Fig 5B). Like the single-cell RNA-seq, we detected a bias of reads with a total of 582 that mapped onto the 5′UTR and 3′LTR regions of the HIV genome. Finally, we aimed to detect chimeric reads, that is, mapping to both the host human and the HIV-1 reference genome, but only detected three chimeric reads.

## Discussion

HIV enters the CNS shortly after infection, targeting microglia and other brain macrophages and playing hide and seek with the immune system (Chaillon et al, 2020). Although ART suppresses HIV replication, the best available regimens do not eradicate replication-competent HIV from the blood and other tissues, including the brain (Cochrane et al, 2022), and HIV is actively transcribed during ART (Dube et al, 2023). The extent to which HIV is actively transcribed during ART is unknown, and the effects of HIV infection on the microglial gene expression are also unclear. The Last Gift program is a unique study that enrolls people with HIV and a terminal illness who are willing to donate their bodies to HIV cure research. The participants agree to a rapid research autopsy procedure within 6 h of death to collect live cells with preserved viability and intact nucleic acid and proteins. This unique resource

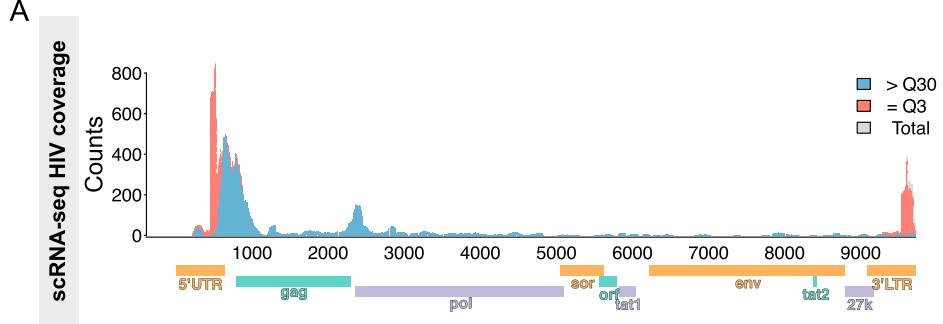

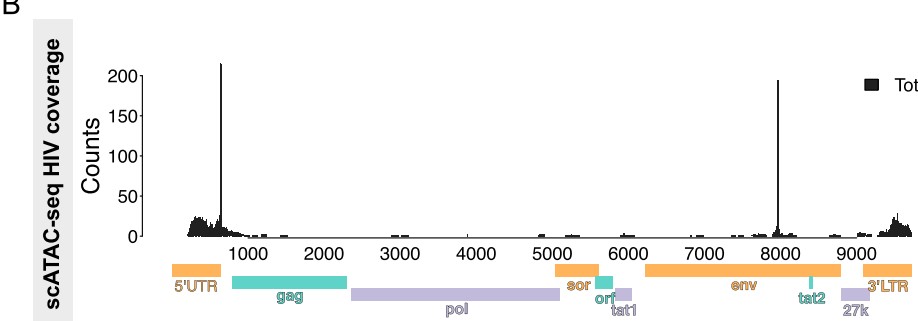

**Figure 5. Alignment track of HIV transcripts across the annotated HXB2 consensus sequence.**
**(A, B)** Integrative Genomics Viewer plots of (A) HIV RNA and (B) HIV DNA in chromatin accessible regions. Read coverage is displayed as a histogram in the top panel.

allowed us to isolate fresh human microglia from the brains of people with HIV while on ART and to perform multi-omics within a very short time frame after death (Perry et al, 2020; Rawlings et al, 2020; Riggs et al, 2022). We generated a comprehensive gene expression atlas and chromatin accessibility landscape of microglia from brains collected from people with HIV.

Here, we performed single-cell transcriptomics on CD45[+] cells isolated from the brains of Last Gift participants on suppressive ART. Based on the gene expression profile, we describe six microglial subpopulations. Compared with homeostatic microglia, which represented most of the CD45[+] cells, microglial subpopulations, including interferon-responsive, activated, proliferative, and inflammatory microglia, showed the reduced expression of homeostatic microglial signature genes, for example, *CX3CR1*, *P2RY12*, and *TMEM119*. Our findings complement previous single-cell RNA-seq studies, which showed similar microglial subpopulations in aging brains (Masuda et al, 2019; Sankowski et al, 2019; Olah et al, 2020). Among these microglial phenotypes with the increased expression of inflammatory (CCL2/3/4, EGR2/3) and interferon-responsive (ISG15, IFIT1/3) profiles (Masuda et al, 2019; Sankowski et al, 2019; Olah et al, 2020; Tansley et al, 2022), as well as subpopulations with increased expression of heat-shock proteins and immediate–early genes, which are similar to microglia isolated from the white matter of patients with multiple sclerosis, we identified microglial phenotypes with increased expression of heat-shock proteins and immediate–early genes (Miedema et al, 2022). We also identified a microglial cluster defined by the increased expression of members of the AP-1 transcription factor family, which may orchestrate the gene expression program to establish an immune response. We did not identify a clear microglial cluster that exhibited a gene expression profile reminiscent of "disease-

associated microglia" that were described in various mouse models, in particular, of neurodegeneration (Keren-Shaul et al, 2017; Gratuze et al, 2023); however, ART may suppress the generation of distinct microglial phenotypes. The identification, or lack of identification, of particular microglial subtypes seen in other systems is likely due to changes resulting from different models, as well as genetic variability and various environmental cues and genetic variability.

Next, we compared the gene expression of HIV RNA[+] microglia to microglia with undetectable HIV RNA. We found enrichment for genes related to interferon signaling and chemokines in HIV RNA[+] cells. Transcriptomic analyses of the brain tissue showed that increased detectable HIV RNA levels correlated with increased interferon signaling, whereas gene expression linked to synaptic and neuronal function was down-regulated (Sanna et al, 2021). We show a down-regulation of *CD4* and up-regulation of *CCR5* and *CXCR4* mRNAs in HIV RNA[+] microglia. These findings support studies demonstrating down-regulation of *CD4* mRNA in HIV-infected T cells (Hoxie et al, 1986; Salmon et al, 1988) and up-regulation of *CCR5* mRNA in infected microglia (Vallat et al, 1998). Using single-cell RNA-seq of the CSF of people with HIV, chemokines such as *CCL8*, *CXCL10*, and *CXCL9* were dysregulated under ART similar to our findings in the brain (Farhadian et al, 2018).

Using scATAC-seq, we present a single-cell chromatin accessibility map of isolated CD45[+] cells. We found DNA motifs enriched for PU.1/ETS family members, which are lineage-determining transcription factors of microglia (Heinz & Glass, 2012; Gosselin et al, 2017). As shown in human fetal microglia, the PU.1/ETS transcription factor family is required for microglial identity (Kracht et al, 2020). Other transcription factors included members of the CEB/P and RUNX families. In contrast to the gene expression profile of

microglia, genome-wide chromatin accessibility was more heterogeneous between the three participants evaluated in this study; we could not assign distinct regions of open chromatin to different microglial states, which is in line with a recently published study on human microglial cell states (Sun et al, 2023). Additional markers, for example, H3K27ac for active promoters and enhancers, may facilitate the definition of active gene regulatory elements, which will allow the identification of transcriptional regulators for these distinct microglial subpopulations.

We detected reads of HIV RNA in ~0.5% of isolated microglia. This aligns with findings showing that replication-competent proviruses likely persist in a small number of microglia and perivascular macrophages as shown by immunostaining and in situ hybridization (Churchill et al, 2006; Thompson et al, 2011; Joseph et al, 2019). Although we did not detect any HIV RNA⁺ perivascular macrophages, we would anticipate there is some infection in this compartment and the lack of detection is most likely due to the low cell numbers obtained. The number of cells we found in which HIV RNA was detected was similar to that in a previous study of people with HIV encephalitis (Plaza-Jennings et al, 2022), but that study did not detect any HIV RNA⁺ cells in people with HIV on ART (Plaza-Jennings et al, 2022). The discrepancy may be due to differences in the technical design, as the single-nucleus RNA-seq used in the prior study is less sensitive than the scRNA-seq approach used here (Thrupp et al, 2020).

The small number of infected cells here is higher than previous studies focusing on the T cell compartment. HIV transcription was detectable only in a small number of CD4⁺ T cells—less than 0.000016% or 16.4 per million cells—of people with HIV with viral suppression on ART (Finzi et al, 1997; Wong et al, 1997), whereas another study suggested an infection frequency of 0.01–0.001% (Siliciano et al, 2003). When a quantitative viral outgrowth assay was used to measure the HIV reservoir, the frequency of latently infected cells in people on ART was estimated at 0.1 and 10 cells per million resting CD4⁺ T cells (Finzi et al, 1997; Siliciano et al, 2003), and probably even less frequently in monocytes (Veenhuis et al, 2023). In the CSF of people with HIV, only a few CD4⁺ T cells were positive for HIV RNA (Farhadian et al, 2018; Suzuki et al, 2022), and detectable HIV RNA was linked to increased neuronal dysfunction as measured by brain magnetic resonance spectroscopy (Suzuki et al, 2022).

Another recent study leveraging the Last Gift cohort detected cell-associated HIV DNA and HIV RNA in isolated microglia from the cortex and basal ganglia. The same study was the first to use a viral outgrowth assay to demonstrate that HIV could be reactivated from the CNS (Tang et al, 2023). Recently, the frontal cortex was nominated as a major brain reservoir of intact HIV DNA in people with HIV under ART and it would be interesting to decipher the genomic landscape of microglia in other brain regions as well (Angelovich et al, 2023). Together, these findings suggest that integrated HIV DNA in microglia may contribute to ongoing disruption of microglia function in the CNS of people with HIV who are virally suppressed and may serve as a viral reservoir.

The number of microglia with HIV DNA integrated into open regions of the host's genome as determined by single-cell ATAC-seq was similar across individuals. The Last Gift participant with the highest detectable level of HIV RNA in bulk tissue also showed the highest number of HIV RNA⁺ and HIV DNA⁺ microglia in the brain. HIV only integrates once in each host cell genome, potentially explaining the small number of detected HIV DNA integration sites into open regions of the host's genome. Only three reads mapped both to the host and to the viral genome, which did not enable us to do an integration site analysis.

A significant limitation of our study is the small number of people with HIV, exclusively male. In addition, this study only included the DLPFC and did not examine other brain regions and surrounding structures such as the meninges and choroid plexus. Although both are significant caveats, they are understandable and necessary given the unique nature of the tissue and the specific limitations of the sample acquisition process. Another limitation of the acquisition process is the loss of spatial resolution, as we were unable to determine the relationships between infected and uninfected immune cells and neighboring cells such as neurons using these types of sequencing analyses. In addition, our isolation protocol and technical limitations of the microfluidic single-cell RNA-seq and single-cell ATAC-seq technology might underestimate the true number of HIV RNA⁺ and HIV DNA⁺ microglia. Recent technological advances in spatial genomics, transcriptomics, and proteomics may help to resolve some of these technical issues, and potentially shed more light on intercellular communications in the brain of people with HIV (Zhuang, 2021; Piwecka et al, 2023). It is likely that concerted efforts between multi-disciplinary research teams at various institutes will be necessary to help to improve our understanding of the immune system and in particular that of tissue-specific resident immune cells in the context of HIV infection (Farber, 2021).

Taken together, using rapid autopsy tissue from the Last Gift study, we show that microglia produce HIV RNA, despite ongoing ART, although the number of microglia with detectable HIV RNA and DNA was low. Our study demonstrated that only a minority of microglia actively produce HIV RNA and HIV DNA in open regions of chromatin. Newer technologies may enable the unbiased detection of HIV-infected cells directly from tissue and may help to further decipher the molecular signatures of immune cells isolated from various distant anatomical niches (Clark et al, 2023). The gene expression profile of HIV RNA⁺ microglia showed up-regulation of interferon-responsive genes. Our data provide a step forward in solidifying the rationale for therapeutic strategies targeting HIV in microglia.

# Materials and Methods

### Last Gift program and rapid research autopsy

The Last Gift program enrolls altruistic, terminally ill people with HIV on ART for close perimortem follow-up. The inclusion criteria for the individuals in this study were as follows: diagnosis of HIV with less than 6 mo to live; on suppressive ART; and no CNS malignancy or immune checkpoint chemotherapy. The study was approved by the IRBs of the UCSD (IRB no. 160563). Written informed consent was obtained from all three participants. DLPFC (Brodmann area 9) specimens were obtained from three Last Gift study participants.

The design of the rapid autopsy protocol allowed us to collect brain tissues within 4 h of death (Chaillon et al, 2020; Rawlings et al, 2020; Wang et al, 2022). Brain samples were immediately processed for microglial isolation for scRNA-seq and scATAC-seq or snap-frozen for ddPCR and H3K27ac ChIP-seq.

### HIV DNA extraction and droplet digital PCR

Genomic DNA was extracted from ~200 mg of BA9 using QIAGEN DNeasy Blood & Tissue Kit (catalog 69504; QIAGEN) according to the manufacturer's protocol. Levels of total HIV DNA were quantified by ddPCR using Bio-Rad QX200 Droplet Reader, and copy numbers were calculated as the mean of three replicate measurements and normalized to one million cells, as determined by the RPP30 assay (Pinheiro et al, 2012; Strain et al, 2013).

### Isolation of CD45$^+$ cells from the human frontal cortex

The human frontal cortex was homogenized by gentle manual dissociation on ice (Gosselin et al, 2017). Briefly, samples were dissociated in an isolation buffer consisting of HBSS without magnesium and calcium (14175-095; Life Technologies), 1% bovine serum albumin, and 1 mM EDTA using a 2 ml polytetra-fluoroethylene douncer and pestle (358026; Wheaton). Resulting cell suspensions were centrifuged and resuspended in 37% isotonic Percoll (P4937; Sigma-Aldrich), and centrifuged at 600 rcf for 30 min at 14°C with no acceleration and deceleration. Pelleted cells were washed twice with isolation buffer and filtered through a 70-$\mu$m strainer (BD Falcon). Cells were then incubated in 1:100 with an Fc receptor blocking antibody (422302; Human TruStain FcX, BioLegend) for 30 min at 4°C, and then with CD11b-PE (301306; BioLegend) or CD11b-APC (101212; BioLegend), and CD45-APC/Cy7 (304014; BioLegend) or CD45-488 (103122; BioLegend). DAPI (Sigma-Aldrich) was used for live/dead staining. After staining, cells were centrifuged, resuspended in an isolation buffer, and kept on ice until FACS.

### FACS of CD45$^+$ cells

CD45$^+$ cells were sorted using Sony SA900 (100 $\mu$m nozzle, purity mode, sample chilling). Flow cytometry data were analyzed using FlowJo software (Becton Dickinson & Company, version 10.8.1). Sorted cells were either immediately processed for scRNA-seq or pelleted and resuspended in FBS with 10% DMSO, and stored at –80°C until processed for scATAC-seq.

### Single-cell RNA-seq on CD45$^+$ cells

The resulting CD45$^+$ cells were centrifuged at 300 rcf for 10 min at 4°C and then resuspended in PBS supplemented with 400 mg/ml bovine serum albumin (B6917; Sigma-Aldrich). Individual cells were coupled to beads using the 10X Genomics Chromium Controller. Single-cell sequencing libraries were prepared following the 10X Genomics protocol for single-cell 3′ using v3.1 chemistry and sequenced using either an Illumina NovaSeq 6000 or HiSeq 4000 on a rapid run.

### Single-cell ATAC-seq on CD45$^+$ cells

Archived CD45$^+$ cells were thawed and counted using a hemocytometer. We targeted 5,000 cells per sample. Libraries were performed according to the manufacturer's instructions (scATAC-seq v1 protocol, 10X Genomics). Cells were permeabilized using lysis buffer (10 mM Tris–HCl, pH 7.4 [15567027; Thermo Fisher Scientific], 10 mM NaCl [ICN15194401; Thermo Fisher Scientific], 3 mM MgCl$_2$ [194698; MP Biomedicals Inc.], 0.1% Tween-20 [P7949; Sigma-Aldrich], 0.1% IGEPAL-CA630 [I8896; Sigma-Aldrich], 0.01% digitonin [G9441; Promega], and 1% BSA in nuclease-free water) and incubated for 5 min on ice. Permeabilized nuclei were washed using washing buffer (lysis buffer without IGEPAL-CA630 and digitonin), centrifuged for 5 min at 500 rcf, and resuspended in nucleus buffer (10X Genomics). An aliquot was mixed with trypan blue and counted using a hemocytometer. Up to 10,000 nuclei were tagmented before loading onto a Chromium Controller; libraries were generated according to the manufacturer's specifications (Chromium Next GEM Single Cell ATAC Library and Gel Bead Kit v1, 1000175; Chromium Next GEM Chip H Single Cell Kit, 1000162; Single Index Kit N Set A, 1000212, 10X Genomics). Libraries were amplified for 10 PCR cycles. SPRIselect reagent (Beckman Coulter) was used for size selection and clean-up steps. Final library concentration was assessed by Qubit dsDNA HS Assay Kit (Thermo Fisher Scientific), and fragment size was inspected using the High Sensitivity D1000 ScreenTape assay on a TapeStation 4200 (Agilent). Libraries were sequenced using an Illumina NovaSeq 6000 using the following read lengths: Read 1: 50 cycles; Read 2: 50 cycles; Index 1: 8 cycles; and Index 2: 16 cycles.

### Nucleus isolation from the frozen tissue

Frontal cortex samples were homogenized in 1% formaldehyde in DPBS (Corning) using a motorized homogenizer (Fisherbrand PowerGen 125), and incubated on a rocker at RT for 10 min. Fixed homogenates were quenched with 0.125 M glycine at RT on a rocker for 5 min. Next, homogenates were centrifuged at 1,100 rcf in a swinging bucket centrifuge. The following steps were all performed on ice except where indicated. Homogenates were washed twice with NF1 buffer (10 mM Tris–HCl, pH 8.0, 1 mM EDTA, 5 mM MgCl$_2$, 0.1 M sucrose, and 0.5% Triton X-100 in ultrapure water) and centrifuged at 1,100 rcf for 5 min at 4°C in a swinging bucket centrifuge. Next, pellets were resuspended in 5 ml NF1 buffer and dounced five times in a 7-ml Wheaton Dounce Tissue Grinder (DWK Life Sciences) using a "loose" pestle. After 30 min of incubation on ice, homogenates were dounced 20 times with a "tight" pestle and filtered through a 70-$\mu$m strainer. To remove myelin debris, homogenates were underlaid with a sucrose cushion (1.2 M sucrose, 1 M Tris–HCl, pH 8.0, 1 mM MgCl$_2$, and 0.1 M DTT) and centrifuged at 3,200 rcf for 30 min with acceleration and brakes on "low." Nucleus pellets were washed with NF1 buffer, centrifuged at 1,600 rcf for 5 min, and stored at –80°C.

### Fluorescence-activated nucleus sorting of PU.1$^+$ nuclei

Pellets of brain nuclei were washed twice in staining buffer (HBSS without magnesium and calcium, 5% BSA, and 1 mM EDTA), then

resuspended in 0.2 ml staining buffer, and incubated overnight at 4°C with the following antibodies: NeuN Alexa Fluor 488 (1:2,500, MAB377; Millipore Sigma) and PU.1 Alexa Fluor 647 (1:100, 658004; BioLegend). The next day, nuclei were washed with staining buffer, passed through a 70-μm strainer, and immediately before the sort stained with 0.5 μg/ml DAPI. Nuclei for the cell type of origin were sorted on a MoFlo Astrios EQ sorter (Beckman Coulter). Sorted nuclei were pelleted in a staining buffer at 1,600 rcf for 10 min and stored at −80°C until processed for H3K27ac ChIP-seq.

### H3K27ac ChIP-seq on PU.1[+] nuclei

Chromatin immunoprecipitation (ChIP) for H3K27ac was performed as previously described (Gosselin et al, 2017; Nott et al, 2019). Fixed, sorted nuclei (~200,000 nuclei per sample) were resuspended in 130 μl ice-cold LB3 (10 mM Tris–HCl, pH 7.5, 100 mM NaCl, 1 mM EDTA, 0.5 mM EGTA, 0.1% Na-deoxycholate, 0.5% N-lauroylsarcosine, and 1 X protease inhibitor cocktail). Chromatin was sheared by sonication using Covaris E220 Focused-ultrasonicator (Covaris) with the following settings: time, 240 s; duty, 5.0; PIP, 140; cycles, 200; amplitude, 0.0; velocity, 0.0; dwell, 0.0. The lysates were adjusted to 250 μl with LB3 and further diluted with 25 μl 10% Triton X-100 (final concentration 1%). Samples were spun down at maximum speed at 4°C for 10 min. For DNA input control, 3 μl of the lysate was taken and volume-adjusted to 25 μl with TT (10 mM Tris–HCl, pH 8, and 0.05% Tween-20), and stored at 4°C until library preparation. For immunoprecipitation, 20 μl of Dynabeads Protein A (10001D; Thermo Fisher Scientific) and H3K27ac antibody (2 μl serum, 39135; Active Motif) were added to the diluted lysates and rotated overnight at 4°C. Beads were collected on a magnet, washed thrice each with wash buffer I (20 mM Tris–HCl, pH 7.5, 150 mM NaCl, 1% Triton X-100, 0.1% SDS, and 2 mM EDTA) and wash buffer III (10 mM Tris–HCl, pH 7.4, 250 mM LiCl, 1% Triton X-100, 0.7% Na-deoxycholate, and 1 mM EDTA), twice with ice-cold TET (10 mM Tris–HCl, pH 7.5, 1 mM EDTA, and 0.2% Tween-20), and once with TE-NaCl (10 mM Tris–HCl, pH 8, 1 mM EDTA, and 50 mM NaCl), and finally resuspended in 25 μl TT. Libraries from ChIP and DNA input samples were prepared with NEBNext Ultra II DNA Library Prep Kit (NEB) reagents according to the manufacturer's protocol on the beads suspended in 25 μl TT (10 mM Tris–HCl, pH 7.5, and 0.05% Tween-20), with reagent volumes reduced by half. Next, DNA was eluted and crosslinks were reversed by adding 4 μl 10% SDS, 4.5 μl 5 M NaCl, 3 μl EDTA, 1 μl proteinase K (20 mg/ml), and 20 μl water, and incubating for 1 h at 55°C, then 30 min overnight at 65°C. DNA was purified using 2 μl of SpeedBeads (GE Healthcare), diluted with 20% PEG8000 and 1.5 M NaCl to a final volume of 12% PEG, and eluted with 12.5 μl TT. DNA contained in the eluate was then amplified for 14 cycles in 25 μl PCRs using NEBNext High-Fidelity 2X PCR Master Mix (NEB) and 0.5 mM each of primers Solexa 1 GA and Solexa 1 GB. The resulting libraries were size-selected by gel excision to 225–350 bp, purified, and single-end–sequenced using an Illumina NovaSeq 6000.

### Single-cell RNA-seq analysis

Sequencing reads were processed with CellRanger version 3.0.1 developed and maintained by 10X Genomics. A custom reference was created by appending the HIV genome (K03455) into hg38 downloaded from Ensembl (GRCh38 v93) along with gene annotations (GTF). The unique molecular identifier (UMI) count matrices labeled with gene symbols and cell barcodes were imported into R (4.0.3) using Seurat (v4.0.2) (Satija et al, 2015). Cells whose detected gene number is less than 200 and genes that are expressed in less than three cells are removed from the following analysis. SCTransform function, which applied the regularized negative binomial regression on 3,000 variable genes, was used per dataset to normalize the scale of the UMI count data (Hafemeister & Satija, 2019). Harmony (Korsunsky et al, 2019) was used to integrate data from three individuals. Dimensionality reduction was performed using principal component analysis. Unbiased clustering was performed with 0.4 resolution by the Louvain algorithm based on the first 30 principal components. Nonlinear dimension reduction using the first 30 principal components was achieved by the Uniform Manifold Approximation and Projection technique for visualization. A total of 25,091 CD45[+] cells (after removal of Cluster 9 and mito content >20%) were analyzed. Marker genes were selected using the Wilcoxon signed-rank test.

### Single-cell ATAC-seq analysis

Sequencing reads of the three single-cell ATAC-seq samples were demultiplexed and processed using the CellRanger software package 3.0.1. Reads were aligned to the human reference hg38 and the HIV-1 reference genome (HXB2, GenBank K03455.1). The fragment files generated by CellRanger were then tagged by read and sample and combined into a unique fragment file. For data analysis, we used the ArchR package (Granja et al, 2021). After the removal of doublets, dimension reduction was performed using Harmony (Korsunsky et al, 2019). Peaks were called using HOMER's findPeaks function in total microglia and microglial subclusters. To identify de novo motifs enriched in peak regions over the GC-matched background, HOMER's motif analysis (findMotifsGenome.pl) was used (Heinz et al, 2010).

### H3K27ac motif analysis

Peaks of H3K27ac were called with HOMER's findPeaks function for each tag directory with -style histone parameter. To quantify H3K27ac signal anchored on ATAC-seq peaks, we used HOMER's annotatePeaks function with "-noadj," "-size 1,000." Next, we filtered annotated H3K27ac peaks with low tag counts below 10. To identify de novo motifs enriched in peak regions over the background, HOMER's motif analysis findMotifsGenome function (-size 200 -len 8, 10, 12) was used. The background peaks used random genome sequences generated automatically by HOMER.

## Data Availability

UCSC Browser tracks for the ATAC-seq and H3K27 ChIP-seq can be accessed via https://genome.ucsc.edu/s/jschlachetzki/Schlachetzki_LastGift_Microglia_20230504. All next-generation sequencing data generated in this study are accessible from

GEO and Array Express via GSE221688. The single-cell RNA-seq data are publicly available via https://apps.bxgenomics.com or specifically via https://apps.bxgenomics.com/bxg/app/scrnaview/app_project_review.php?ID=1040 (scATAC) and https://apps.bxgenomics.com/bxg/app/scrnaview/app_project_review.php?ID=1039 (scRNA).

## Supplementary Information

## Acknowledgements

We would like to thank the participants of the Last Gift study for their tissue donation and support. This study was supported by NIH grants 1RF1AG061060-01 (CK Glass), 1R21MH134401-01 (JCM Schlachetzki and X Yang), K99HD111686 (X Yang), and R01AI164559 (HOPE Delaney Collaboratory). This work was also supported by the Department of Veterans Affairs, the James B Pendleton Charitable Trust, the NIH-funded San Diego Center for AIDS Research (P30 AI036214), and grants from the National Institutes of Health AI036214 (DM Smith—CFAR), AI169609 (BEAT Delaney Collaboratory), DA055491 (A Chaillon), MH062512 (HNRC), UL1TR001442 (ACTRI), AI164559 (HOPE Delaney Collaboratory), and U24 MH100928 (California NeuroAIDS Tissue Network).

### Author Contributions

JCM Schlachetzki: conceptualization, formal analysis, funding acquisition, investigation, methodology, and writing—original draft, review, and editing.

S Gianella: resources, funding acquisition, investigation, and writing—review and editing.

Z Ouyang: data curation, software, formal analysis, and writing—review and editing.

AJ Lana: data curation, investigation, and writing—review and editing.

X Yang: investigation, methodology, and writing—review and editing.

S O'Brien: investigation and writing—review and editing.

JF Challacombe: formal analysis.

PJ Gaskill: data curation, methodology, and writing—review and editing.

KL Jordan-Sciutto: data curation, investigation, and writing—review and editing.

A Chaillon: formal analysis, funding acquisition, and writing—review and editing.

D Moore: investigation and writing—review and editing.

CL Achim: investigation and writing—review and editing.

RJ Ellis: investigation and writing—review and editing.

DM Smith: resources and funding acquisition.

CK Glass: resources, supervision, funding acquisition, and writing—review and editing.

### Conflict of Interest Statement

The authors declare that they have no conflict of interest.

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
