## [Reviewer comments · Life Science Alliance]

Life Science Alliance

Gene expression and chromatin conformation of microglia in virally suppressed people with HIV

Johannes Schlachetzki, Sara Gianella, Zhengyu Ouyang, Addison Lana, Xiaoxu Yang, Sydney O'Brien, Jean Challacombe, Peter Gaskill, Kelly Jordan-Sciutto, Antoine Chaillon, David Moore, Cristian Achim, Ronald Ellis, Davey Smith, and Christopher Glass

DOI: <https://doi.org/10.26508/lsa.202402736>

Corresponding author(s): Christopher Glass, University of California - San Diego School of Medicine and Johannes Schlachetzki, University of California, San Diego

Review Timeline:

Submission Date:	2024-03-25
Editorial Decision:	2024-05-28
Revision Received:	2024-07-04
Editorial Decision:	2024-07-08
Revision Received:	2024-07-14
Accepted:	2024-07-15

Transaction Report:

May 28, 2024

Re: Life Science Alliance manuscript #LSA-2024-02736-T

Dr. Christopher K Glass
University of California San Diego Medical Center
Division Cell and Molecular Medicine University of California - San Diego 9500 Gilman Dr
La Jolla, CA 92093-0651

Dear Dr. Glass,

Thank you for submitting your manuscript entitled "Gene expression profiles and chromatin conformation of microglia in virally suppressed people with HIV" to Life Science Alliance. The manuscript was assessed by expert reviewers, whose comments are appended to this letter. We invite you to submit a revised manuscript addressing the Reviewer comments.

Thank you for this interesting contribution to Life Science Alliance. We are looking forward to receiving your revised manuscript.

Sincerely,

B. MANUSCRIPT ORGANIZATION AND FORMATTING:

Reviewer #1 (Comments to the Authors (Required)):

Given that the brain is a putative reservoir for HIV-1, the authors leveraged the Last Gift Cohort to investigate variation of microglia in the DLPFC based on transcriptomic signatures and the chromatin accessibility landscape to inform and potentially correlate gene expression data that on open and hence, transcriptionally-active regions. The authors do not overstate the conclusions of this observational study conducted on three male brains from individuals with very different co-morbid conditions. Namely, they identified 15 different CD45+ clusters and based on a limited set of myeloid specific genes attempted to parse out the six microglia clusters, from perivascular macrophages, monocytes and neutrophils. Alignment of sequence data with a reference HIV genome was used to identify cells harboring HIV DNA and RNA. Previous relevant studies are cited and this study adds to the field and shows what insights are possible using human brain tissue and the significant limitations to be overcome in future experimental designs.

The contributions of all the authors to the study as presented is not clear.

Reviewer #2 (Comments to the Authors (Required)):

In "Gene expression profiles and chromatin conformation of microglia in virally suppressed people with HIV" , Schlachetzki et al. Isolated viable microglia from post-mortem brain tissue of 3 donors with HIV that were on antiviral combination therapy until shortly before they passed away. They then subjected these cells to single-cell RNA sequencing as well as scATACseq to detect any possible presence of HIV virus, either expressed as RNA, or integrated in the genome. Using the scRNA data they further analyzed the effects of HIV on the expression pattern of infected microglia. The number of replicates in the dataset is understandably low, with data available from 3 donors. Overall, the quality of the data seems high and given the unique nature of the tissue, these data still represents a valuable resource.

There are a few points that could use some clarification prior to publishing:

1. In figure 4 you analyse the gene expression differences of the microglia cells that are positive for HIV contrasted to a random selection of microglia cells from the population. The majority of the HIV positive cells are assigned to clusters that are not homeostatic microglia but overall express markers of cellular stress, inflammation or activation. Therefore, the fact that these gene expression differences are found could merely reflect the expression signature of that cluster, rather than an expression pattern induced by HIV. In these clusters, there are many cells with a similar pattern that are not HIV positive. So are the genes induced by HIV? Or are the cells that show this pattern more likely to be infected? And are there any gene expression changes when you compare HIV positive and negative cells from the same cluster?
2. In the discussion on page 11 you write a section on defective-proviruses being able to transcribe HIV RNA. Then how you detect HIV RNA in 0.005% of microglia that you go on to assign to coming from replication-competent provirus. How do you know what type of virus (defective or replication-competent) was present in these microglia?
3. In the discussion it is written that: Additional markers, e.g., H3K27ac for active promoters and enhancers... The methods section, which includes headings: Nuclei isolation from frozen tissue, FACS sorting of PU.1+ nuclei and H3K27ac ChIPseq on PU.1+ nuclei, suggests that you already performed this analysis. What was the outcome and can this be included in the manuscript?

Minor points:

1. The image of the FACS plot depicted in figure 1c does not seem to represent an actual sample. Given that almost all cells in this figure are both CD45 and CD11B positive, this had me confused initially. In the text you mention that all CD45+ cells were isolated. This statement made more sense after seeing fig S1, where it is obvious that in some donors there is also a proportion of cells that are CD45+, but CD11B-. I would recommend to include an actual FACS plot of one of the samples in the main figure as well.
2. In the text on page 5 it is written that 15 microglia clusters were identified, while there are 16 (starting from 0).
3. The manuscript does not consistently follow guidelines for nomenclature of gene names, in the discussion on page 11 some gene names are not italicized.
4. On page 11 it is written: We show a downregulation of CD4+ and... This seems like a mistake. Should it be CD4?
5. On page 11: It is estimated that roughly 25-30%... what does this refer to? Is it 25-30% of cells?

Reviewer #1 (Comments to the Authors (Required)):

Given that the brain is a putative reservoir for HIV-1, the authors leveraged the Last Gift Cohort to investigate variation of microglia in the DLPFC based on transcriptomic signatures and the chromatin accessibility landscape to inform and potentially correlate gene expression data that on open and hence, transcriptionally-active regions. The authors do not overstate the conclusions of this observational study conducted on three male brains from individuals with very different co-morbid conditions. Namely, they identified 15 different CD45+ clusters and based on a limited set of myeloid specific genes attempted to parse out the six microglia clusters, from perivascular macrophages, monocytes and neutrophils. Alignment of sequence data with a reference HIV genome was used to identify cells harboring HIV DNA and RNA. Previous relevant studies are cited and this study adds to the field and shows what insights are possible using human brain tissue and the significant limitations to be overcome in future experimental designs.

Response: We would like to thank Reviewer 1 for the positive comments on our manuscript.

The contributions of all the authors to the study as presented is not clear.

Response: We now added the following paragraph to the Acknowledgment section outlining the contributions of all the authors to the study.

“JCMS conceived and designed the study, processed tissue to isolate microglia, analyzed multi-omics data, interpreted the results, wrote the first draft of the manuscript. SG co-directs the Last Gift Program and performed all rapid autopsies with her team, interpreted the data, edited the manuscript. ZO processed scRNA-seq data and performed quality control of all single cell data. AJL collected samples, performed microglia isolation and library preparations, edited the manuscript. XY interpreted the single cell RNA-seq data, edited the manuscript. SMO performed library preparations, edited the manuscript. JFC processed scATAC-seq data. PJG interpreted the data, edited the manuscript. KLJS interpreted the data, edited the manuscript. AC helped with rapid autopsy procedures analyzed multi-omics data, edited the manuscript. DM collected clinical data, edited the manuscript. CLA interpreted the data, edited the manuscript. RJE interpreted the data, edited the manuscript. DMS co-directs of the Last Gift Program, helped with rapid autopsy procedures, interpreted the data, edited the manuscript. CKG supervised the project, interpreted the data, edited the manuscript.”

Reviewer #2 (Comments to the Authors (Required)):

In "Gene expression profiles and chromatin conformation of microglia in virally suppressed people with HIV" , Schlachetzki et al. Isolated viable microglia from post-mortem brain tissue of 3 donors with HIV that were on antiviral combination therapy until shortly before they passed away. They then subjected these cells to single-cell RNA sequencing as well as scATACseq to detect any possible presence of HIV virus, either expressed as RNA, or integrated in the genome. Using the scRNA data they further analyzed the effects of HIV on the expression pattern of infected microglia. The number of replicates in the dataset is understandably low, with data available from 3 donors. Overall, the quality of the data seems high and given the unique nature of the tissue, these data still represents a valuable resource.

Response: We would like to thank Reviewer 2 for appreciating our work.

There are a few points that could use some clarification prior to publishing:

1. In figure 4 you analyse the gene expression differences of the microglia cells that are positive for HIV contrasted to a random selection of microglia cells from the population. The majority of the HIV positive cells are assigned to clusters that are not homeostatic microglia but overall express markers of cellular stress, inflammation or activation. Therefore, the fact that these gene expression differences are found could merely reflect the expression signature of that cluster, rather than an expression pattern induced by HIV. In these clusters, there are many cells with a similar pattern that are not HIV positive. So are the genes induced by HIV? Or are the cells that show this pattern more likely to be infected? And are there any gene expression changes when you compare HIV positive and negative cells from the same cluster?

Response: We agree that the majority of HIV-positive cells reside in clusters that are not representative of homeostatic microglia. This raises the possibility that the observed differences may reflect the general activation state of these clusters rather than a direct effect of HIV infection. As suggested by the reviewer, we reanalyzed our data and

compared HIV RNA positive and negative cells from the same cluster. We found that the gene expression profiles of microglia with detectable HIV RNA are very similar to those of microglia with no detectable HIV RNA (Fig. 1 for reviewers). Specifically, we performed differential gene expression analysis of HIV RNA+ microglia (Cluster 1: 19 cells, Cluster 3: 19 cells, Cluster 4: 26 cells, Cluster 7: 14 cells) and compared them to randomly selected similar cell numbers from the same cluster. No significant changes in gene expression were found in Clusters 3 and 7, and only 6 and 1 genes were significantly altered in Clusters 1 and 4, respectively.

Fig. 1: Differential gene expression analysis within in clusters between HIV-RNA+ microglia and microglia without detectable HIV-RNA.

2. In the discussion on page 11 you write a section on defective-proviruses being able to transcribe HIV RNA. Then how you detect HIV RNA in 0.005% of microglia that you go on to assign to coming from replication-competent provirus. How do you know what type of virus (defective or replication-competent) was present in these microglia?

Response: The reviewer is correct; we cannot distinguish whether the HIV RNA detected in microglia originates from defective or replication-competent proviruses. Both types of viruses can contribute to the immune response, but our current methods do not allow us to differentiate between them in this context. We recognize the importance of this distinction and its implications for understanding the role of HIV-infected microglia in the immune response and viral persistence. However, due to the limitations of our current detection techniques, we are unable to address this differentiation in our study. Future research utilizing more advanced methodologies may be able to provide clearer insights into the specific contributions of defective versus replication-competent HIV in microglia. We now removed this section from the discussion.

3. In the discussion it is written that: Additional markers, e.g., H3K27ac for active promoters and enhancers... The methods section, which includes headings: Nuclei isolation from frozen tissue, FANS sorting of PU.1+ nuclei and H3K27ac ChIPseq on

PU.1+ nuclei, suggests that you already performed this analysis. What was the outcome and can this be included in the manuscript?

Response: We performed H3K27ac analysis on PU.1 isolated nuclei from the three Last Gift participants. We now included motif analysis and browser session for this data set (Figure 3E/F) and included an annotated peak list as supplementary data 3.

Minor points:

1. The image of the FACS plot depicted in figure 1c does not seem to represent an actual sample. Given that almost all cells in this figure are both CD45 and CD11B positive, this had me confused initially. In the text you mention that all CD45+ cells were isolated. This statement made more sense after seeing fig S1, where it is obvious that in some donors there is also a proportion of cells that are CD45+, but CD11B-. I would recommend to include an actual FACS plot of one of the samples in the main figure as well.

Response: We now removed the original plot in Figure 1c with an actual FACS plot depicted in Suppl. Figure 1.

2. In the text on page 5 it is written that 15 microglia clusters were identified, while there are 16 (starting from 0).

Response: We apologize for not making it clear that one cluster was removed from the analysis due to low quality. We have clarified this in the methods section.

3. The manuscript does not consistently follow guidelines for nomenclature of gene names, in the discussion on page 11 some gene names are not italicized.

Response: Thank you for pointing this out. We have now looked through the manuscript to remove inconsistency in the nomenclature of gene names. We used nomenclature as suggested by HNGC.

4. On page 11 it is written: We show a downregulation of CD4+ and... This seems like a mistake. Should it be CD4?

Response: Agreed, we corrected this.

5. On page 11: It is estimated that roughly 25-30%... what does this refer to? Is it 25-30% of cells?

Response: We agree that this statement was not very precise. In light of the previous comment by the reviewer that we cannot distinguish between defective or replication-

competent HIV viruses, we have removed these sentences from the discussion on page 11 to provide a more succinct and clear discussion. Removed sentences from the manuscript: Although effective ART prevents the full replication cycle, current ART regimens do not entirely prevent the transcription of HIV genes. It is estimated that roughly 25-30% - including those harboring defective proviruses- transcribe HIV RNA even during suppressive ART.

July 8, 2024

RE: Life Science Alliance Manuscript #LSA-2024-02736-TR

Dr. Christopher K Glass
University of California, San Diego
Division Cell and Molecular Medicine University of California - San Diego 9500 Gilman Dr
La Jolla, CA 92093-0651

Dear Dr. Glass,

Thank you for submitting your revised manuscript entitled "Gene expression and chromatin conformation of microglia in virally suppressed people with HIV". We would be happy to publish your paper in Life Science Alliance pending final revisions necessary to meet our formatting guidelines.

- please be sure that the authorship listing and order is correct
- please add ORCID ID for corresponding author--you should have received instructions on how to do so
- please add the Twitter handle of your host institute/organization as well as your own or/and one of the authors in our system
- title in the manuscript and submission page must match
- please consult our manuscript preparation guidelines <https://www.life-science-alliance.org/manuscript-prep> and make sure your manuscript sections are in the correct order
- please add a Conflict of Interest statement to your main manuscript text
- please use the [10 author names, et al.] format in your references (i.e. limit the author names to the first 10)
- please add an Author Contributions section to your main manuscript text
- contributions listed for authors Peter Gaskill, Kelly Jordan-Sciotto and Antoine Chaillon, in the system, do not qualify for authorship. Please either update the contributions in our system and in the Author Contributions section of the manuscript, or let us know if any author should be removed.
- please remove figures from main manuscript file, they should only be uploaded separately. Figure legends should be placed in the manuscript file after the Reference list
- Table 1 can be left in the main manuscript file, or uploaded separately
- please add a callout for Figures 2A-C and S1A-C to your main manuscript text. Each section of each figure needs to be called out
- please indicate that written informed consent was obtained from the patients

LSA now encourages authors to provide a 30-60 second video where the study is briefly explained. We will use these videos on social media to promote the published paper and the presenting author (for examples, see <https://docs.google.com/document/d/1-UWCfbE4pGcDdcgzcmiuJl2XMBJnxKYeqRvLLrLSo8s/edit?usp=sharing>). Corresponding or first-authors are welcome to submit the video. Please submit only one video per manuscript. The video can be emailed to contact@life-science-alliance.org

A. FINAL FILES:

-- Summary blurb (enter in submission system): A short text summarizing in a single sentence the study (max. 200 characters including spaces). This text is used in conjunction with the titles of papers, hence should be informative and complementary to

the title. It should describe the context and significance of the findings for a general readership; it should be written in the present tense and refer to the work in the third person. Author names should not be mentioned.

B. MANUSCRIPT ORGANIZATION AND FORMATTING:

Sincerely,

July 15, 2024

RE: Life Science Alliance Manuscript #LSA-2024-02736-TRR

Dr. Christopher K Glass
University of California - San Diego School of Medicine
Division Cell and Molecular Medicine University of California
San Diego 9500 Gilman Dr
La Jolla, CA 92093-0651

Dear Dr. Glass,

Thank you for submitting your Resource entitled "Gene expression and chromatin conformation of microglia in virally suppressed people with HIV". It is a pleasure to let you know that your manuscript is now accepted for publication in Life Science Alliance. Congratulations on this interesting work.

DISTRIBUTION OF MATERIALS:

Again, congratulations on a very nice paper. I hope you found the review process to be constructive and are pleased with how the manuscript was handled editorially. We look forward to future exciting submissions from your lab.

Sincerely,
